# Management of Sjogren’s Dry Eye Disease—Advances in Ocular Drug Delivery Offering a New Hope

**DOI:** 10.3390/pharmaceutics15010147

**Published:** 2022-12-31

**Authors:** Kevin Y. Wu, Wei T. Chen, Y-Kim Chu-Bédard, Gauri Patel, Simon D. Tran

**Affiliations:** 1Department of Surgery-Division of Ophthalmology, University of Sherbrooke, Sherbrooke, QC J1G 2E8, Canada; 2Margaret Cochran Corbin VA Campus, New York, NY 10010, USA; 3Faculty of Medicine, University of Sherbrooke, Sherbrooke, QC J1G 2E8, Canada; 4Faculty of Medicine, University of Montreal, Montreal, QC H3T 1J4, Canada; 5Faculty of Dental Medicine and Oral Health Sciences, McGill University, Montreal, QC H3A 1G1, Canada

**Keywords:** sjögren syndrome, dry eye, ocular drug delivery, topical administration, nanocarriers, subconjunctival, episcleral and intravitreal implant, controlled release systems, basic research, pathophysiology

## Abstract

Sjögren’s syndrome is a chronic and insidious autoimmune disease characterized by lymphocyte infiltration of exocrine glands. Patients typically present with dry eye, dry mouth, and other systemic manifestations. Currently, the available molecules and drug-delivery systems for the treatment of Sjögren’s syndrome dry eye (SSDE) have limited efficacy since they are not specific to SSDE but to dry eye disease (DED) in general. The current treatment modalities are based on a trial-and-error approach using primarily topical agents. However, this approach gives time for the vicious cycle of DED to develop which eventually causes permanent damage to the lacrimal functional unit. Thus, there is a need for more individualized, specific, and effective treatment modalities for SSDE. The purpose of this article is to describe the current conventional SSDE treatment modalities and to expose new advances in ocular drug delivery for treating SSDE. A literature review of the pre-clinical and clinical studies published between 2016 and 2022 was conducted. Our current understanding of SSDE pathophysiology combined with advances in ocular drug delivery and novel therapeutics will allow the translation of innovative molecular therapeutics from the bench to the bedside.

## 1. Introduction

The observation of a series of patients with dry eyes, dry mouth, and musculoskeletal pain was described in 1933 in a thesis by the Swedish ophthalmologist Henrik Sjögren. Sjögren’s syndrome (SS) is a chronic and insidious autoimmune disease characterised by lymphocyte infiltration of exocrine glands, typically the lacrimal and salivary glands causing the cardinal signs of xerophthalmia and xerostomia [1].

Global prevalence of SS is currently estimated at 0.06%. Over 90% of affected individuals are female. As much as 10% of patients suffering from dry eye disease (DED) are diagnosed with SS. However, two-thirds of these patients remain undiagnosed, and a median diagnostic delay of 10 years is reported [2]. Diagnosis of SS remains challenging due to the variable course of this condition and wide spectrum of non-specific clinical manifestations. Ocular symptoms include foreign body sensation, burning sensation, eyestrain, photophobia, red eyes, and blurred vision. Symptoms are aggravated with prolonged visual effort (e.g., reading, screentime) and by environmental extremes (e.g., low humidity, extreme cold). However, patients may be asymptomatic or mildly symptomatic despite signs of significant ocular inflammation. Poorly treated SSDE may result in vision-threatening complications such as neurotrophic keratitis, resulting in corneal thinning, ulceration, melt and perforation [3].

The current management for Sjögren’s syndrome dry eye (SSDE) has demonstrated limited efficacy for this condition since it remains non-specific to the disease process and only targets ocular surface inflammation and dryness. A trial-and-error therapeutic approach is generally adopted by ophthalmologists; however, it gives time for the vicious cycle of DED to take place, which eventually causes permanent damage to the lacrimal functional unit [4]. Moreover, topical treatments heavily rely on patients’ physical capacity to administer eyedrops and patient compliance. These shortcomings highlight the need for individualized and specific treatments addressing SSDE’s pathophysiology. The purpose of this article is to describe the current conventional SSDE treatment modalities and expose new advances in ocular drug delivery for treating SSDE.

## 2. Conventional Treatments

Topical administration of molecules remains the cornerstone of drug delivery system for the treatment of SSDE. Occasionally, a topical treatment can be combined with an interventional treatment (e.g., punctal plugs) to increase the bioavailability of topical drugs. At present, systemic administration of molecules is rarely indicated as first-line treatment for SSDE. It is generally used to treat extraocular manifestations (e.g., xerostomia) or systemic manifestations (e.g., pulmonary and renal complications) of SS, but not SSDE alone. However, new molecular targets may support the use of systemic drug delivery in the treatment of SSDE (refer to Section 3 on Novel Treatment Modalities and Advances in Drug-Delivery Systems). Table 1 highlights the mechanisms and treatment consideration of each therapeutic option discussed in this section.

### 2.1. Topical

#### 2.1.1. Artificial Tear Substitutes

Desiccating stress causes epithelial damage which incites an inflammatory cascade. Volume replacement with artificial tears, gels, ointments, or inserts reduce the friction and mechanical disruption between the lid and the globe.

Lubricants are composed of demulcents that lubricate the ocular surface and emollients that slow tear evaporation. There are several types of topical lubricants available with little evidence preferring one over the other, even less so for SSDE [5]. Some studies show that formulations containing hydroxypropyl methylcellulose and isotonic hyaluronate with stronger mucoadhesive properties are preferred for SSDE [6]. There is strong evidence that lubricants with polyacrylic acid are superior to those with polyvinyl alcohol [7]. Non-preserved artificial tears should be highly considered because common preservatives such as benzalkonium chloride can cause toxicity to the ocular surface [5].

Artificial tears are safe and effective for patients with dry eye but frequent dosing for patients with SSDE, who often have chronic and moderate or severe disease, poses as a limitation. Higher viscous formulations such as ointments increase contact time and reduce the frequency of dosing, but can cause blurred vision [7].

#### 2.1.2. Biologic Tear Substitutes

Biologic tear substitutes such as blood-derived autologous serums or platelet rich plasma eyedrops contain nutrients and epitheliotropic factors that are absent in artificial lubricants. They contain epidermal growth factor, fibronectin, platelet derived growth factor-AB, and transforming growth factor-β1, which have anti-inflammatory properties that regulate wound healing and regeneration [8,9]. They are recommended for severe DED. There is strong evidence that platelet rich plasma eyedrops are effective and superior to artificial tears in SSDE [10,11]. The preparation, cost, storage requirements, and frequent dosing of biologics limit its widespread use [12].

#### 2.1.3. Glucocorticoids

SSDE show an increased level of proinflammatory markers in the tear film compared to non-Sjogren’s syndrome dry eye (NSSDE), so treatments targeting inflammation may be implemented earlier in these patients [13]. Topical glucocorticoids are a fast-acting immunosuppressive agent that helps to break the inflammatory cycle of dry eye [14]. Steroid therapy is generally limited to two-week pulses for acute symptoms, or as pre-treatment for delayed onset therapies such as topical cyclosporin. Lower dose formulations such as 0.1% fluorometholone or 0.5% loteprednol have demonstrated safety and efficacy in patients with severe SSDE [15].

#### 2.1.4. Cyclosporin A

Topical Cyclosporine A is an immunomodulatory drug with anti-inflammatory properties. Cyclosporine is a calcineurin inhibitor that inhibits T-cell activation and prevents the release of proinflammatory cytokines, including IL-2 and IL-4 [16]. It prevents T-cell mediated apoptosis, which increases goblet cell density and may promote mucin production and tear flow [17]. In mouse models, it has been shown to decrease expression of proinflammatory cytokines and chemokines IL-1β, TNF-α, IL-6, intercellular adhesion molecule 1, and vascular cell adhesion molecule [18]. It is a safe and effective treatment that is recommended in mild-moderate stage dry eye.

Cyclosporine requires consistent dosing and has a delayed onset of 4 to 12 weeks but shows a faster response in SSDE. It is as effective as 0.1% fluorometholone with marginally worse side effects including short term ocular pain and conjunctival congestion but has a long term safety profile [19]. It is currently available as Restasis 0.05% and Ikervis 0.1%. Recently, the US FDA approved Cequa 0.09%, a nanomicellar topical formulation of cyclosporine. Nanomicelles are amphiphilic surfactants. They can contain the hydrophobic cyclosporine A and help with its dissolution and penetration through the tear film and other ocular structures. This formulation was shown to safely and significantly improve signs and symptoms of dry eyes [20]. Other ocular delivery systems such as nanomicelle inserts, gels and contact-lens delivery systems are being studied and discussed in the next section (Section 3) [20,21,22].

#### 2.1.5. Lifitegrast

Lifitegrast is an integrin antagonist with immunomodulatory and anti-inflammatory properties, which targets the inflammation in SSDE. It binds to lymphocyte function associated-1 found naturally on helper T-cells and prevents their binding to the intercellular adhesion molecule-1 ligand on vascular endothelial cells, thus preventing activation of the inflammatory cascade and subsequent cytokine release. It is recommended for mild to moderate dry eye. Studies support its safety and efficacy. It has a faster time to onset and shows improvement in 2–4 weeks. There are currently no studies comparing the effectivity of lifitegrast to cyclosporine, or evaluating its effectivity in SSDE [23]. It causes more frequent but nonserious side effects, such as pain, irritation, and dysgeusia lasting up to 4 h [24].

#### 2.1.6. Diquafosol Sodium

3% Diquafosol eye drops are approved for the treatment of dry eye in Japan. It binds to the P2Y2 receptor found on the ocular surface to stimulate conjunctival epithelial and goblet cell secretion, significantly increasing tear volume versus controls [25]. It also improves meibomian gland dysfunction. A meta-analysis of 9 randomized control trials found possible effectivity in dry eye. A study on SSDE demonstrated effectivity, though patients were permitted to use topical corticosteroids upon worsening symptoms [26].

### 2.2. Systemic

#### 2.2.1. Muscarinic Agonists

Oral muscarinic agonists such as pilocarpine and cevimeline are FDA approved for oral dryness in patients with SS, and are used off-label for SSDE [4]. They bind M1 and M3 agonists in exocrine glands, which leads to increased secretory function. Oral pilocarpine for 3 months showed improvement of some but not all dry eye signs. Increased sweating is a common side effect, with increase urinary frequency or flushing less frequently seen. Cevimeline is an alternative to pilocarpine but is associated with significantly higher rate of nausea and sweating [27].

#### 2.2.2. Hydroxychloroquine

Commonly used as an anti-rheumatic treatment, hydroxychloroquine is thought to exert immunomodulatory effects as well. The efficacy of hydroxychloroquine in SSDE has not been established. A 2021 systematic review and meta-analysis showed improvement in xerostomia, but did not show significant relief of ocular or other systemic symptoms [28]. Common side effects include retinal toxicity, with a risk of cardiac, gastrointestinal, dermatological, and neuromuscular adverse effects [29].

### 2.3. Interventional Procedures

#### 2.3.1. Punctal Occlusion

Punctal occlusion with collagen or silicone plugs can be performed quickly and safely in office to reduce tear drainage and improve aqueous retention on the tear film. Punctal plugs may last at least 16 weeks, unless spontaneously extruded. Liu et al. found significant improvement in markers for aqueous and evaporative dry eye. Patients with NSDDE showed better improvement. Accumulation of inflammatory cytokines on the ocular surface may exacerbate meibomian gland dysfunction [30]. Other studies have found that punctal plugs were not more effective than artificial tears in Sjogren’s patients [31]. Common side effects are plug extrusion, infection, or inflammation.

#### 2.3.2. Amniotic Membrane

Amniotic membranes are derived from the inner layer of the placenta and provide mechanical support, increase wound healing growth factors, and reduce pro-inflammatory cytokines [32]. A series of case reports have noted success in treatment of signs and symptoms of SSDE when applied to the ocular surface [33]. The main adverse events are foreign body sensation and blurred vision.

#### 2.3.3. Neurostimulator

Endogenous tear production can be stimulated through chemical, mechanical, or electric stimulation of the nasal trigeminal nerve at the nasal mucosa [34]. Daily application of TrueTear, a now discontinued electric intranasal tear stimulator, showed immediate increased tear production in patients with SDDE and NSSDE [35]. iTear, a newer electromechanical external tear stimulator, stimulates the external nasal nerve with high sonic frequency and significantly improves Schirmer score with a rare occurrence of dizziness and headaches [36]. Varenicline is a twice daily intranasal spray FDA approved in 2021 that chemically activates the nasolacrimal reflex. It is a nicotinic acetylcholine receptor agonist that binds to the nasociliary nerve within the nasal cavity. Four weeks of treatment saw improved results at week 12 [36,37]. Pharmacy-sponsored studies show significant improvement after 4 weeks of treatment and showed significantly better results compared to lifitegrast. The most common adverse events are mild transient sneezing and cough.

#### 2.3.4. Cul-de-Sac Ophthalmic Insert

Cul-de-sac ophthalmic inserts are non-surgically placed inserts that can be used as drug-delivery systems. An existing product is Lacrisert, a cul-de-sac ophthalmic insert containing 5 mg of hydroxypropyl cellulose which is approved for moderate to severe dry eyes, including keratoconjunctivitis sicca (KCS). It works by stabilizing, thickening and prolonging tear film break up time. The insert can be applied by the patient and does not require professional insertion. The efficacy and tolerability of the implants in the treatment of SSDE have been observed since the 1980’s and continues to be an option in treatment of dry eyes [38].

#### 2.3.5. Subcutaneous Reservoir

In patients with severe dry eye, mechanical dacryoreservoirs can be implanted to provide continuous lubrication drop. The device is implanted subcutaneously at the anterior abdominal wall and connected to a catheter that releases lubrication at the upper conjunctival fornix. It is reserved for advanced cases due to risk of infections requiring removal of the device [39].

#### 2.3.6. Submandibular Salivary Gland Transplantation

Submandibular salivary gland transplantation into the temporal fossa replaces the tear film in patients with severe aqueous deficiency dry eye. It is contraindicated in patients with salivary gland dysfunction or pathology such as in SS [40,41]. Post-operative complications include epiphora and microcystic corneal edema.

**Table 1 pharmaceutics-15-00147-t001:** Summary of conventional treatment options of SSDE.

Accepted Treatment Options of SSDE		
	Mechanism of Action	Advantages & Considerations
Artificial lubricants	Tear replacement via artificial tears, ointments, inserts to reduce epithelial damage [5]	High frequency of dosing Preservatives may induce toxicity [5] Ointments may induce blur [7]
Biologic Tear Substitutes Autologous Serum (AT) Platelet rich preparations (PRP)	Tear replacement and wound repair through epitheliotropic and neurotrophic factors [8,9]	PRP may be superior to AT [10,11] Cost, preparation, and storage requirements are limitations [12] High frequency of dosing [8]
Topical Corticosteroids	Immunosuppression through NF-κB suppression and phospholipase A inhibition [14]	Fast acting [14] Risk of ocular hypertension, glaucoma, cataract, infection [15]
Topical Cyclosporine A	Immunomodulation and anti-inflammation via calcineurin inhibition [16]	Long term safety, site specific side effects Delayed onset up to 3 months [19]
Topical Lifitegrast	Immunomodulation and anti-inflammation through LFA-1/ICAM inhibition [23]	Systemic side effects including dysgeusia, headaches [23] Delayed onset up to 1 month [23]
Topical Diquafosol Sodium	Tear volume stimulation through P2Y2 stimulation [25]	May have added benefit with meibomian gland dysfunction [25] Not available in the USA [25]
Pilocarpine and Cevimeline	Secretagogues by cholinergic agonists [4,27]	Effective for both oral and ocular symptoms [4]. Off label for dry eye. Common systemic side effects include sweating, increased urinary frequency, and flushing [4,27]
Punctal Occlusion	Improve aqueous retention by reducing tear drainage [30]	May be less effective in Sjogren’s dry eye [31]. Surface inflammation may exacerbate signs [30] Spontaneous extrusion, inflammation, and infection [30]
Amniotic Membrane Graft	Promote wound healing by mechanical support, growth factor delivery, and anti-inflammation [32]	Foreign body sensation, blurred vision, discomfort [32]
Extranasal iTear Stimulator	Tear volume stimulator by electromechanical nasolacrimal stimulation through exterior of nose [35]	Highly effective with low side effects including dizziness and headaches [35,36]
Intranasal Varencicline	Tear volume stimulator by chemical nicotinic acetylcholine receptor neurostimulation at the nasociliary nerve [36,37]	Bypasses topical administration Nonserious side effects such as transient sneezing, cough [36,37]
Not utilized		
Hydroxychloroquine	Immunomodulation, anti-inflammation [28]	Retinal and multisystem toxicity [29]
Submandibular Salivary Gland Transplant	Tear volume restoration [40,41]	Contraindicated in glandular disease such as Sjogren’s [41,42]

## 3. Novel Treatment Modalities and Advances in Drug-Delivery Systems

Despite the current advances in clinical pharmacology and extensive research aiming to improve ocular manifestations in SS patients, treating SSDE remains a challenge for ophthalmologists. Our literature review revealed several novel treatment modalities on the horizon. Some of these use advanced drug-delivery systems to increase the efficacy and tolerability of existing therapies. Others aim to target the autoimmune pathways and pathogenesis of SSDE. Table 2 highlights the mechanisms and treatment consideration of each therapeutic option discussed in this article.

### 3.1. Immunomodulators

Current treatments for SSDE are only aimed to relieve and manage ocular surface inflammation and dryness. Systemic immunomodulators and disease modifying treatments have been studied in hopes of targeting the underlying pathogenesis and autoimmunity of Sjögren’s syndrome.

B-cell targeted therapies are the most studied, with the largest number of patients and most reported randomized control trials. Rituxumab is a monoclonal antibody targeting CD20 antigen that facilitates B-cell depletion with secondary T-cell regulatory responses. Results of studies on the efficacy of rituximab are variable and its use is controversial [42]. Belimumab is a B-cell activating factor (BAFF) inhibitor that may improve disease markers and dryness symptoms, though the available evidence is weak. Headaches and transient neutropenia are common, with one serious report of pneumococcus meningitis [43]. Epratuzumab targets CD22, a co-receptor of B-cell receptor. An old study of 16 patients in 2006 showed positive responses, but no studies have been conducted since [44].

Recent studies of other treatments have shown possible benefit in SSDE. Ianalumab is another BAFF receptor inhibitor in which recent studies have shown reduced disease activity [45]. Iguratimod is a macrophage migration inhibitory factor (MIF) that downregulates proinflammatory cytokines and rebalances Th1, TH17, and T-reg. A systematic review showed improvement in symptoms, inflammatory markers, and salivary and lacrimal secretory functions in Sjogren’s patients who were taking concurrent conventional immunosuppressive therapy [46,47]. Abatacept is a T-cell costimulation modulator, targeting CD28-CD80 and CD86, preventing T-cell activation. In a multi-center, double-blind, placebo-controlled trial of 187 patients with active moderate to severe SS, no significant improvement was found, but there was clear impact on SS related biomarkers, such as CXCL13 [48]. In contrast, a single center randomized double-blind, placebo-controlled study showed improvement in symptoms, dry eye tests, and laboratory markers [49]. Further studies of the safety and efficacy of these treatments should be considered.

Several other agents have been studied in SS, but not necessarily in relation to dry eye. This includes anti-inflammatory agents infliximab and etanercept, which target TNF-a, and anakinra, an IL-1 antagonist [50,51]. Tocilizumab, an IL-6 receptor agonist, showed no impact on immunoglobulins, complement, or systemic disease activity, which suggested that IL-6 may not be a main contributor to peripheral B-cell activation in SS [52]. Janus kinase-1 (JAK-1) inhibitor filgotinib and spleen tyrosine kinase (STK) inhibitor lanraplenib, which are approved in Japan and Europe, as well Bruton’s tyrosine kinase (BTK) inhibitor tirabrutinib that is currently being studied, have also shown no effect in patients with Sjogren’s [53]. Baminercept, inhibitor of lymphotoxin-b receptor fusion protein, failed to improve disease [54].

Many other clinical trials are underway. RSLV-132 is an RNAse fusion protein that was aimed to reduce IFN activity. In phase II clinical trials on 30 patients, it showed improvement in severe fatigue in patients with SS, but unexpectedly upregulated IFN-inducible genes instead. Common adverse events were fatigue and non-serious infections, with one reported unrelated hospitalization [55]. Further studies are needed.

### 3.2. Anterior Segment Ocular Drug-Delivery Technologies

#### 3.2.1. TOP1360

TOP1360 is a non-systemic kinase inhibitor (NSKI) currently undergoing phase 2 clinical trials for the treatment of dry eye disease. Interest for NSKI in DED stems from recent studies which have shown the potential of NSKI to reduce inflammation in diseases such as ulcerative colitis, chronic obstructive pulmonary disease and arthritis. These agents decrease inflammation by targeting kinases involved in innate and adaptive immunity signaling, including p38-alpha, Syk, Srx and Lck. These kinases have been shown to be upregulated in DED. Taylor et al. evaluated TOP1360 0.1% ophthalmic solution against placebo in 61 adults with reported history of DED in both eyes for at least 6 months. This treatment was shown to be equivalent to placebo in terms of safety and tolerability. While the primary endpoint was not evaluation of TOP1360 efficacy in treatment of DED, significant improvement was observed by day 15 of treatment in patients receiving TOP1360, with patients reporting relief in ocular dryness, ocular discomfort, foreign body discomfort and grittiness [56].

#### 3.2.2. Lacritin

Lacritin is an endogenous tear glycoprotein which is observed to promote basal tearing and maintain ocular surface integrity by targeting SDC1 expressed on corneal and conjunctival epithelia via heparinase cleavage. This results in a signal cascade which promotes cell secretion and proliferation. When comparing tears from 15 SS human patient to tears from 14 healthy human controls, Vijmasi et al. found that lacritin active fragments were significantly lower in SS tears [57]. They further studied the therapeutic potential of lacritin vs. placebo in treatment of KCS in mice eyes. Within 1 week of treatment, lacritin-treated eyes showed a 32% increase in tearing (*p* ˂ 0.001) and by week 3, tearing had increased to 46% (*p* = 0.01) and corneal epithelial damage was shown to be significantly decreased. Additionally, while lymphocytic infiltration of lacrimal glands remained similar in lacritin-treated eyes and untreated eyes, lymphocyte foci per millimeter square are of lacrimal gland tissues were decreased significantly. Lymphocyte foci expansion is a sign of exocrine gland autoimmune disease. Therefore, lacritin could be hypothesized to modulate lacrimal gland inflammation by protecting against focal lymphocyte infiltration.

#### 3.2.3. RGN-259

RGN-259 ophthalmic solution is a synthetic of the naturally occurring thymosin β4, a protein found in almost all cells and promotes wound repair in multiple tissues. It has been studied in the treatment of neurotrophic keratitis non-responsive to anti-inflammatory agents. It promotes ocular surface healing, increases corneal epithelial cell migration, and reduces corneal pro-inflammatory cytokines. The safety and efficacy of 0.1% Thymosine β4 twice daily has been established in phase II clinical trials [58]. Studies in mouse models showed increased tear production, mucin, and goblet cell density, as well as decreased anti-inflammatory factors, at least as well as diquafosol, cyclosporine A, and lifitegrast [59]. Two phase III clinical trials were conducted: ARISE-2 did not meet its primary outcome but noted significant improvement in symptoms, and ARISE-3 found clinically significant improvement after one and two weeks of treatment. There are plans for an additional clinical trial, ARISE-4, in 2024. Excellent safety profile was reported.

#### 3.2.4. Contact Lens Drug-Delivery

Contact lenses have been studied as vehicles for drug theory because of their effect of drug release kinetics and their physical barrier properties. Their use allows for increased drug contact time and proximity to the cornea, resulting in the highest drug bioavailability when compared to other non-invasive formulations [60]. Different drug loading techniques and formulations have been studied which are uniquely adapted to accommodate loading of either hydrophobic or hydrophilic molecules. Simple soaking is a straightforward drug-loading technique that involves submerging the lens into a drug-containing solution. However, drug uptake and release are limited by lens factors (including thickness, water-content, etc.), and drug properties (solubility, molecular weight, etc.). To improve drug-uptake and extend release time, vitamin E-coated lenses can be used with the soaking technique. The addition of vitamin E creates a diffusion barrier for the drug. To better accommodate for hydrophobic drugs, surfactants can be added to existing contact lenses. This allows for the formation of micelles which encourage preferential partition of hydrophobic molecules, such as cyclosporine A. Studies also looked at the use of colloidal particles containing active ingredients which can be used to coat lenses. Successful sustained delivery of dexamethasone was observed with this technique. The most recent drug-loading technique is molecular imprinting through which a specific molecular meshwork is created to fit a particular molecule. This technique has been applied to diclofenac with observed sustained release of therapeutic doses.

Contact-lens drug delivery with existing commercial lens materials has been evaluated with select active pharmaceutical products which have shown efficacy in dry-eyes.

##### Contact Lens Drug-Delivery—Anti-Inflammatory

Cyclosporine A, also known as Restasis, is an FDA-approved topical formulation used for the treatment of dry eyes. Peng et al. observed a delivery rate of 15 days when loading cyclosporine A on a silicone hydrogel lens. This rate was further increased to 20 days by the addition of surfactant molecules to the lens mixture [61].

Dexamethasone-delivery from contact-lenses has also been studied. Boone et al. observed that dexamethasone-loading onto silicone hydrogel lenses did not provide adequate drug release time. Addition of surfactant, soaking in vitamin E, changes in monomer ratios and use of drug-loaded colloidal particles were attempted with improvement in drug release time up to 3 months [62].

Diclofenac was also successfully loaded onto contact lenses by different methods including monomer addition and novel molecular imprinting techniques. These allowed for sustained in vitro release of up to 2 weeks. The molecular imprinted contact lenses further showed release of diclofenac comparable to the maximin dose delivered by commercial eye drops [63].

##### Contact Lens Drug-Delivery—Secretagogue

Secretagogue work by stimulating the secretion of tear fluid and mucins. Three families of secretagogue exist: dinucleotides, cholinergic agonists, and mucin secretagogue. While secretagogues have been successfully applied as eye drops for treatment of dry eyes, contact lens drug delivery has only been studied with dinucleotides. Studies in rabbit eyes showed that diadenosine tetraphosphate when delivered by contact lens provided tear secretion above baseline for up to 360 min [63].

##### Contact Lens Drug-Delivery—Osmoprotectant

Osmolarity plays an important role in the pathophysiology of dry eyes. In that context, osmoprotectants have been studied and available data suggest that osmoprotectant solutions may be beneficial in dry eye treatment. Contact lenses have been tried as a possible drug delivery system for these compounds which achieved drug release times of up to 10 h. However, further studies to determine the clinical impact of extended-release osmoprotectants on dry eyes are necessary [63].

##### Contact Lens Drug-Delivery—Rewetting and Comfort Agents

Rewetting and comfort agents are known common symptomatic treatments for dry eyes. These include acrylic comfort agents and polysaccharide comfort agents. The studies related to their delivery via contact lens systems resulted in the approval and addition to the market of Focus Dailies with AquaComfort, a polyvinyl alcohol-containing single use contact lens [63].

Hydroxypropyl methylcellulose is one of the polysaccharide comfort agents which has been observed to significantly improve corneal staining score and break up time in SSDE. Contact-lens drug delivery of hydroxypropyl methylcellulose was also studied and showed encouraging results. Imprinted contact lenses resulted in a 6-fold increase in delivery rate of hydroxypropyl methylcellulose when compared to the eye drop formulation [63].

While research is promising, successful demonstration of significantly higher safety and efficacy over conventional eye drops is lacking. Furthermore, despite the different available techniques for drug-loading onto contact lenses, some new materials have yet to undergo extensive testing to establish their applicability to contact lens production.

#### 3.2.5. Subconjunctival and Episcleral Implants

Subconjunctival or episcleral implants have many advantages including sustained drug delivery locally and better patient compliance. Existing implants include Surodex containing dexamethasone used to treat inflammation post-cataract surgery and LX201 containing cyclosporine A, which is currently undergoing phase III studies for prevention of corneal graft rejection [60].

There is evidence that cyclosporine A subconjunctival implants may be used in the treatment of KCS. Studies conducted on dogs and mares have shown encouraging results. Barachetti et al. used 30% cyclosporine implants in 15 dogs with KCS, which were well tolerated and contributed significantly to reducing conjunctival hyperemia, corneal neovascularization, corneal opacity, and ocular discharge [64]. Mackenzie et al. treated an 8-year-old mare with a 1-month history of KCS using a cyclosporine implant and topical cyclosporine. The mare showed improved tear-production and absence of clinical signs 9 days post-op [65]. Therefore, subconjunctival and episcleral implants could provide routes of investigations for future management of KCS.

#### 3.2.6. Ocular Iontophoresis

Ocular iontophoresis is a drug delivery technique utilizing mild electric charges to cross into anterior and posterior segments. It allows for higher bioavailability and reduced clearance of active ingredients when compared with topical eye drops and has better compliance than ocular injections. Use of iontophoresis for administration of iodide and dexamethasone have both been studied in the treatment of dry eyes.

In a prospective study including 28 patients, iodide iontophoresis was compared to iodide alone in the treatment of dry eyes. Patients having received iodine iontophoresis maintained significant objective and subjective improvement of dry eyes at 3 months, while the control group only maintained subjective improvement at 1 month with no significant improvement maintained at 3 months [66]. Iontophoresis may in this case play a role in prolonging treatment benefits in dry eye patients.

Ocular iontophoresis was also applied to dexamethasone in a single-center, double-masked, randomized, placebo-controlled phase II trial which enrolled 102 patients with confirmed signs and symptoms of dry eyes. Patients were randomized to either the placebo, lower dose iontophoresis, or higher dose iontophoresis group. While significant improvements were not observed for clinical signs, significant improvement was noted in symptoms for the lower dose iontophoresis group when compared to placebo [67].

While iontophoresis drug delivery has shown favorable effects in improving symptoms of dry eyes, further studies need to be done to determine specific dosages and applicability to other active compounds.

#### 3.2.7. Colloidal Nanocarriers for Anterior Segment Disorders

Colloidal nanocarriers are a product of new developments in nanotechnology which could allow for improved pharmacokinetics of existing ocular drugs by their ability to shield drugs from enzymatic destruction and allow passage through ocular barriers, consequently increasing bioavailability. Several nanocarrier subtypes exist.

##### Colloidal Nanocarriers for Anterior Segment Disorders—Nanomicelles

Nanomicelles are amphiphilic surfactants that can contain hydrophobic drugs such as cyclosporine A. It can be used in a topical eye drop formulation. As mentioned in Section 2.3 of this article, the recently FDA approved Cequa, a nanomicellar topical formulation of cyclosporine, was shown to safely and significantly improve signs and symptoms of dry eyes [20]. Its pharmacokinetic advantages were identified as improved residence time in ocular tissues and higher concentration in ocular tissues when compared to original formulations of the same compound (Restasis). Pharmacokinetic studies reported that concentrations of cyclosporine delivered by nanomicelle formulation were 3.6 and 3.44-fold higher in conjunctival and scleral tissues, respectively, when compared to Restasis [20]. Nanomicelles are currently being analyzed for delivery of nucleic acids including miRNA.

##### Colloidal Nanocarriers for Anterior Segment Disorders—Nanoparticles

Nanoparticles are 50–500 nm nanocarriers which have been studied to carry lipophilic drugs, hydrophilic drugs, polynucleotides, and even gene therapy. Research on nanoparticles loaded with active pharmaceutical products studied in the treatment of SS dry eyes have been conducted. De Campos et al. evaluated chitosan nanoparticles as a possible vehicle for cyclosporine A delivery to the ocular surface in rabbits. This study found that therapeutic concentration was achieved for at least 48 h in targeted corneal and conjunctival tissues without significant systemic levels [68].

##### Colloidal Nanocarriers for Anterior Segment Disorders—Liposomes

Liposomes are colloidal nanocarriers with the ability to encapsulate both hydrophilic and hydrophobic compounds. Liposome encapsulation of azithromycin has been studied for the treatment of dry eyes in rats with higher efficacy in reducing symptoms when compared to hyaluronic acid drops. While topical liposomes have been studied with other ocular drugs, studies remain to be done on their application to SSDE treatments [69].

##### Colloidal Nanocarriers for Anterior Segment Disorders—Dendrimers

Dendrimers are nanocarriers which provide the advantage of customizability. Their shape, size, and surface functional groups can be adjusted to accommodate different drug molecules and target tissues. While no research has been done on their application to dry-eye treatments, active products such as dexamethasone have been successfully cross-linked to dendrimers and studied in rat models for treatment of corneal inflammation, diabetic retinopathy(DR), and age related macular degeneration(AMD). These studies reported enhanced ocular permeability when compared to the eye drop formulations [70,71].

##### Colloidal Nanocarriers for Anterior Segment Disorders—Microneedles

Microneedles are an existing drug delivery vehicle which is used for transdermal drug delivery. It is a minimally invasive technique in which microneedles coated with either hydrophilic or hydrophobic drugs are applied to anterior or posterior ocular segments. This formulation has been shown to penetrate the corneal layer and allow for localized drug delivery. Further studies need to be done to evaluate their potential clinical use [72].

##### Colloidal Nanocarriers for Anterior Segment Disorders—Nanowafers

Nanowafers are small disc formulations which are intended to be smeared on the ocular surface with a fingertip. These discs are engineered for continuous release of drugs from the anterior ocular surface, improving residence time and bioavailability and acting as a protective layer for the corneal surface. Their use with dexamethasone in the control of inflammation in mice dry eyes was evaluated. 24 h of sustained release of dexamethasone was reported following an initial burst in the first hour. The dexamethasone nanowafers also showed equivalent efficacy to the drops in reducing expression of inflammation markers on the cornea [69].

Colloidal nanocarrier research has had encouraging results which could be applied to treatment of SSDE. The ability to have sustained and localized drug delivery systems could have a beneficial effect on drug bioavailability, efficacy, and safety.

### 3.3. Posterior Segment Ocular Drug-Delivery Technologies

#### Intravitreal Implants

Intravitreal implants are drug eluting devices inserted in the vitreous humor that can be used for prolonged drug release in ocular disorders including DR,, diabetic macular edema (DME), AMD, central retinal vein occlusion (CRVO), and posterior uveitis.

Implants having received FDA approval include Durasert technology system, NOVADUR, and I-vation TA [60]. While these implants have yet to be investigated in the treatment of dry eye syndrome, their use in other disorders may indicate a potential avenue for use in dry eye treatment.

The approved formulations of Durasert includes implants loaded with ganciclovir for the treatment of cytomegalovirus retinitis and with fluocinolone acetonide for the treatment of both posterior uveitis and diabetic macular edema [73]. The Durasert fluocinolone-loaded insert Iluvien is being further studied for efficacy in age related macular degeneration and macular edema due to retinal vein occlusion in comparison to Lucentis [60].

There is some evidence that the use of fluocinolone acetonide implants may be beneficial in Sjogren’s syndrome-related keratopathy. One case report presented a patient with Sjogren syndrome and multiple corneal perforations, who received treatment in one eye with Iluvien implants. While the untreated eye required further keratoplasty and amniotic membrane transplants, the treated eye did not require further surgical intervention in the 6 months following implantation of Iluvien [74].

Therefore, intravitreal implants represent a new potential drug delivery technology that could be beneficial in the treatment of dry-eye syndromes.

**Table 2 pharmaceutics-15-00147-t002:** Summary of emerging treatment options for SSDE.

Clinical Trials		
Biologic Immunomodulators	Immunomodulation through innate and adaptive immune system. Rituxumab: anti-CD20 antibody [42] Belimumab: anti-BAFF antibody [43] Epratuzumab: anti-CD20 antibody [44] Ianalumab: anti-BAFF receptor antibody [45] Iguratimod: MIF inhibitor [46,47] Abatacept: CD80/86:CD28 costimulation inhibitor [48,49]	Limited studies on efficacy and safety with often conflicting results [42,43,44,45,46,47,48,49].
RSLV-132	RNAse targeting IFN [55]	Improved severe fatigue in Sjogren’s [55]. Non-serious side effects like fatigue and infections [55].
RGN-259	Anti-inflammatory, anti-apoptotic, and wound healing properties through thymosine β4 activity [58]	Reported excellent safety profile [58,59] Pending additional dry eye trial (ARISE-4) in 2023 [59].
Mesenchymal Stem Cells and Exosomes	Immunomodulation, regeneration	May restore gland function. Exosomes are more accessible. Stem cells limited by donor compatibility and preparation.
Dexamethasone Subconjonctival Implant (Surodex)	Immunosuppression through NF-κB suppression and phospholipase A inhibition [60]	Sustained drug delivery locally Better patient compliance Currently undergoing phase III studies for prevention of corneal graft rejection.
Cyclosporine A Subconjunctival Implants	Immunomodulation and anti-inflammation via calcineurin inhibition [64,65]	Studies showing encouraging results in animals [64,65]
Iodide Iontophoresis	Use of mild electric charges applied to drug molecules allowing for passage into anterior and posterior segments [66,67]	Higher bioavailability Reduced clearance of active ingredients Better compliance than ocular injection Further studies needed to determine specific dosage and applicability to other active pharmaceutical compounds
Dexamethasone Iontophoresis		
Chitosan Nanoparticles Containing Cyclosporine A	Immunomodulation and anti-inflammation via calcineurin inhibition [68]	Sustained therapeutic drug concentrations in corneal and conjunctival tissues
Azithromycin Liposomes	Immunomodulatory and anti-inflammatory effects through decreased NF-κB, IL-6, IL-8, and MMP-2 activity, and increased TGF-β1	Reduced symptoms of dry eyes in rats Studies remain to be done in SS
Dendrimer Drug-Delivery	Nanocarriers that can be customized to accommodate different drug molecules [70,71]	No research has been done applying to dry-eye treatment Successful cross-linkage of active products used in dry-eye treatment
Nanowafers	Nanodisc formulation intended to be smeared on ocular surface with fingertips [60,69]	Continuous drug release to anterior ocular surface Improved residence time and bioavailability Studied to reduce inflammation markers of cornea in mice.
Fluocinolone Acetonide Intravitreal Implant	Drug-eluting device inserted into vitreous humor [74]	Prolonged drug release Existing technology having received FDA approval. One case report of use for treatment of SS
Not utilized		
Immunomodulators	Infliximab: TNF-a inhibitor [50] Etanercept: TNF-a inhibitor [50] Anakinra: IL-1 antagonist [51] Tocilizumab: IL-6 receptor agonist [52] Filgotinib: JAK-1 inhibitor [53] Ianraplenib: STK inhibitor [53] Tirabrutinib: BTK inhibitor [53] Baminercept: Lymphotoxin-b receptor fusion protein inhibitor [54]	Found ineffective in Sjogren’s disease [50,51,52,53,54]
Microneedle Drug-Delivery	Application of microneedle coated with active ingredients to anterior or posterior ocular segments [72]	Further studies needed to establish potential clinical use

## 4. On the Horizon: Mesenchymal Stem Cells Therapy and Exosomes

Mesenchymal stem cells (MSC) are cells isolated from bone marrow, fatty tissues, umbilical cord, and gingiva, which can differentiate into specialized cells and exert immunoregulatory and regenerative effects. They are studied in patients with autoimmune diseases, including Sjogren’s syndrome, and may be administered intravenously, intraperitoneally, and sometimes intraglandularly [75]. Several studies have demonstrated MSC’s immunomodulatory abilities on T cells, B cells, dendritic cells, and natural killer cells [76]. In both Sjogren’s mouse and human models, treatment with various MSCs have improved salivary flow, reduced lymphocyte infiltration, and reduced inflammatory cytokines [75]. Modulation of Tim-3, an inhibitory receptor, by MSCs have shown alleviation of SS related pathological changes [77].

Different stem cells lines have been shown to reduce proinflammatory cytokines, exhibit anti-apoptotic effects, and rebalance Treg/Th17 cells, which are associated with improving Sjogren’s pathology. Dental pulp-derived stem cells may improve Sjogren’s hyposecretion through the TGF-b/Smad pathway, and studies suggest it may be more effective than bone marrow MSCs [78,79]. Dental follicle MSC decreased inflammatory cell deposits and fibrosis in glandular tissue, and showed differentiation into glandular epithelial cells [80]. Human exfoliated deciduous teeth stem cells, which are from the cranial neural crest, secrete soluble programmed cell death ligand 1 (PD-L1) and alleviate gland inflammation and dryness in mice [81]. Umbilical MSCs exert a local and adaptive immunosuppressive response through CD4^+^FJoxp3^+^ T cells [82]. Other studies also found it also suppressed Vγ4^+^IL-17^+^ that contributed to SS. Regulation of these cells increased saliva flow and reduced lung pathology [83].

However, MSC administration was found to be limited by donor compatibility and long preparation times of autologous transplants [84]. In this case, exosomes derived from MSCs have been shown to overcome these difficulties. Exosomes are a subtype of cell-derived membranous structure called extracellular vesicles, which exists in all biological fluids and contribute to immune response, cardiovascular diseases, central nervous system diseases and cancer. The contribution to physiological process is mainly based on the enclosed proteins within exosomes, including signaling proteins, metabolic enzymes, and antigens. These components are acquired when exosomes are released from the parental cell by budding and take on some of its membrane characteristics including lipid components, proteins, and nucleic acids. Because of their ability to express different types of surface molecules, exosomes can interact with many specific cells, and cross through difficult barriers. Furthermore, the fatty acids components of exosomes confer stability and structural rigidity which contribute to long circulation times in the body. Because of the characteristics mentioned above, exosomes can be used as drug delivery systems [85].

When transplanted or infused in mice, exosomes derived from MSCs were shown to decrease symptoms of SS and differentiate into different salivary gland cells, resulting in restored function of the glands [84]. Labial gland derived MSCs and their associated exosomes have been found to rebalance the expression of miRNA-125b-5p and miRNA-155-5p in CD4+ T cells [86,87]. miRNA-125b inhibits PRDM1 translation, which regulates B-cell differentiation, and miRNA-155 is associated with cytokine production and CD8+T cell proliferation [86,88]. Olfactory ecto-MSC derived exosomes secrete IL-6, which could activate the JAK2/STAT pathway of myeloid-derived suppressor cells to significantly improve saliva flow rate and reduce tissue damage in Sjogren’s mice models [89]. Some studies also observed significant enrichment of miRNA miR-21 by MSC exosomes, which was shown to play a significant role in exosome-related immune regulation [84]. While MSC-derived exosomes show promise in the treatment of SS dry eyes, further studies need to be conducted to better understand the mechanism of action of exosomes and the manufacturing constraint related to their production.

## 5. Conclusions

As of today, the management of SSDE remains a challenge. Current treatment modalities are largely based on therapies used for NSSDE. Novel therapeutic modalities utilizing advanced ocular drugs-delivery systems, such as subconjunctival, episcleral and intravitreal implants, nanocarriers, as well as other controlled release systems, are on the horizon. Moreover, our understanding of the diversity of biological processes underlying the pathogenesis of SSDE benefits the development of new effective agents, specifically targeting the pathophysiology of SS. Distinct submolecular mechanisms underlie therapies that may hold the potential to optimize or revolutionize medical therapeutics. Translating innovative molecular therapeutics from bench to bedside will require extensive clinical trials, but it will present a promising prospect to improve the quality of life of patients suffering from SSDE.

## Data Availability

Not applicable.

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
