# Peer review of "Management of Sjogren’s Dry Eye Disease—Advances in Ocular Drug Delivery Offering a New Hope"

_pharmaceutics, 2022, doi:10.3390/pharmaceutics15010147_

Round 1

Reviewer 1 Report

This article very clearly delineates the current and upcoming treatment strategies for treating ocular Sjögren’s

This is pretty novel interma of the type of review articles I have personally seen and will add an excellent base for anyone who would like to start new research in this field.

No changes recommended.

Author Response

Thank you for your feedback. We greatly appreciate you taking the time to review our article. The treatment of SSDE is an exciting field, and we hope to encourage more researchers to invest their time in investigating new treatment options for this disease. 

Reviewer 2 Report

Dear Authors,

This paper is an interesting work, well written and documented. The article extensively describes the current classical SSDE treatment options as well as the new opportunities in ocular drug delivery for SSDe treatment.

The information could be very useful for the treatment perspectives of the SSDE rare disease (the global prelevance about 0.06%), the treatment of the disease remaining challenging.

I have some remarks as follows:

1. Please add in Table 1 the references.

2. Same observation for Table 2.

3. Please be careful to the sub-sub-chapter numbering. See page 11: “2.4.5. Contact Lens Drug-Delivery”; page 12: “3.2.2.1. Contact Lens Drug-Delivery - Anti-inflammatory” ……; page 13: “3.2.3. Subconjuctival and Episcleral Implants”….

4. Please replace utilized with used, and utilising with using.

5. Other minor English revision and spelling corrections are required.

Author Response

Thank you for your suggestions and comments. We greatly appreciate your time.

1. The references for both Table 1 and Table 2 were added. 

2. The references for both Table 1 and Table 2 are added. 

3. The chapter numbering was corrected. 

4. The word “utilized” and “utilizing” were replaced by “used” and “using” respectively.

5. Minor English mistakes were corrected

Reviewer 3 Report

While the subject of potential new therapies for Sjogrens disease is treated in a comprehensive manner, the title is inconsistent with the review of current therapy. I suggest this section be significantly reduced, perhaps in the form of the included figure.

Author Response

Thank you for your suggestion. We understand your concern. Therefore we decide to broaden the title of our article: “Management of Sjogren's Dry Eye Disease - Advances in Ocular Drug Delivery Offering a New Hope”. We believe that this title is now more consistent with the review of current therapy. We would like to keep the section on current therapy if possible since it gives some foundations to our readers for transitioning to the section on future therapeutic modalities.